# N-Acetylcysteine Alleviates D-Galactose-Induced Injury of Ovarian Granulosa Cells in Female Rabbits by Regulating the PI3K/Akt/mTOR Signaling Pathway

**DOI:** 10.3390/antiox13040384

**Published:** 2024-03-22

**Authors:** Jiawei Cai, Yunpeng Li, Bohao Zhao, Zhiyuan Bao, Jiali Li, Shaoning Sun, Yang Chen, Xinsheng Wu

**Affiliations:** 1College of Animal Science and Technology, Yangzhou University, Yangzhou 225009, China; dx120230166@stu.yzu.edu.cn (J.C.); mx120230883@stu.yzu.edu.cn (Y.L.); bhzhao@yzu.edu.cn (B.Z.); dx120210143@stu.yzu.edu.cn (Z.B.); dx120200140@stu.yzu.edu.cn (J.L.); mz120221510@stu.yzu.edu.cn (S.S.); 2Joint International Research Laboratory of Agriculture & Agri-Product Safety, Yangzhou University, Yangzhou 225009, China

**Keywords:** D-galactose, ovarian granulosa cells, apoptosis, N-acetylcysteine (NAC), PI3K/Akt/mTOR signaling pathway

## Abstract

The ovary plays a crucial role in the reproductive system of female animals. Ovarian problems such as ovarian insufficiency, premature aging, polycystic ovary syndrome, and ovarian cysts may lead to ovulation disorders, abnormal hormone secretion, or luteal dysfunction, thereby increasing the risk of infertility and abortion. Only when the ovarian function and other organs in the reproductive system remain healthy and work normally can female animals be ensured to carry out reproductive activities regularly, improve the pregnancy rate and litter size, promote the healthy development of the fetus, and then improve their economic value. The follicle, as the functional unit of the ovary, is composed of theca cells, granulosa cells (GCs), and oocytes. GCs are the largest cell population and main functional unit in follicles and provide the necessary nutrients for the growth and development of follicles. N-acetylcysteine (NAC) is a prevalent and cell-permeable antioxidant molecule that effectively prevents apoptosis and promotes cellular survival. Over the past few years, its function in boosting reproductive performance in animals at the cellular level has been widely acknowledged. However, its specific role and mechanism in influencing GCs is yet to be fully understood. The objective of this study was to examine the effects of NAC on ovarian damage in female rabbits. For this purpose, D-galactose (D-gal) was first used to establish a model of damaged GCs, with exposure to 1.5 mg/mL of D-gal leading to substantial damage. Subsequently, varying concentrations of NAC were introduced to determine the precise mechanism through which it influences cell damage. Based on the results of the Cell Counting Kit-8 assay, flow cytometry, and Western blotting, it was found that 0.5 mg/mL of NAC could significantly suppress cell apoptosis and promote proliferation. In particular, it decreased the expression levels of Bax, p53, and Caspase-9 genes, while concurrently upregulating the expression of the BCL-2 gene. Moreover, NAC was found to alleviate intracellular oxidative stress, suppress the discharge of mitochondrial Cytochrome c, and boost the enzymatic activities of CAT (Catalase), GSH (Glutathione), and SOD (Superoxide dismutase). RNA sequencing analysis subsequently underscored the critical role of the PI3K/Akt/mTOR pathway in governing proliferation and apoptosis within GCs. These findings demonstrated that NAC could significantly influence gene expression within this pathway, thereby clarifying the exact relationship between the PI3K/Akt/mTOR signaling cascade and the underlying cellular processes controlling proliferation and apoptosis. In conclusion, NAC can reduce the expression of Bax, p53, and Caspase-9 genes, inhibit the apoptosis of GCs, improve cell viability, and resist D-gal-induced oxidative stress by increasing the activity of CAT, GSH, and SOD. The molecular mechanism of NAC in alleviating D-gal-induced ovarian GC injury in female rabbits by regulating the PI3K/Akt/mTOR signaling pathway provides experimental evidence for the effect of NAC on animal reproductive function at the cellular level.

## 1. Introduction

The ovary of female animals serves both as the main reproductive endocrine organ and the site for the production of mature oocytes, which are essential for fertility [1]. The basic functional unit of the mammalian ovary is the follicle, whose development involves a complex physiological process governed by various regulatory substances and hormonal factors. Within the follicles, granulosa cells (GCs) [2], often referred to as follicular somatic cells, play a key role in supporting the development and maturation of oocytes as well as in transmitting endocrine signals [3]. In female rabbits, maintaining a balance between the proliferation and apoptosis of GCs is crucial for proper follicular maturation and development. GCs regulate follicular atresia by participating in the apoptosis process and subsequently modulating the growth and maturation of follicles by secreting cytokines, hormones (such as estrogen and progesterone), and proteins [4,5].

D-galactose (D-gal), a reducing sugar, plays a physiological role when present at suitable concentrations in the body. However, its excessive accumulation can also give rise to increased production of reactive oxygen species (ROS), which subsequently reduce the activity of vital antioxidant enzymes, including CAT, SOD, and GSH [6]. In addition, this oxidative stress often causes metabolic dysfunction, leading to damage and loss of function of biological macromolecules such as lipids, proteins, and DNA. At the same time, ROS activates the inflammatory signaling pathway, accelerates telomere shortening, limits the number of cell divisions, and induces cell senescence. These processes work together to accelerate the aging process of cells and the whole body [7,8]. In this context, studies have shown that D-gal could be used to establish aging models in different cell types, including astrocytes, lens epithelial cells, as well as mesenchymal stem cells [9,10,11].

N-acetylcysteine (NAC) is a widely used cell-permeable antioxidant with a significant role in regulating hormone secretion, improving oocyte quality and aiding ovarian function recovery [12]. Existing research indicates that NAC can improve follicular development and restore ovarian and uterine functions by reducing oxidative stress as well as apoptosis in GCs [13]. Similarly, studies involving mouse models have shown that short-term administration of NAC as an antioxidant for two months could delay the aging process of ovarian oocytes [14,15] by effectively impeding telomere shortening and chromosome instability caused by oxidative stress. This, in turn, promoted oocyte and early embryo development [16,17]. In in vitro experiments, the addition of 1.5 mM of NAC to oocyte maturation medium has been shown to increase the proportion of embryos reaching the blastocyst stage [18]. Moreover, 200 μM of NAC could activate the cAMP and Wnt signaling pathways to significantly enhance proliferation and hormonal secretion by goat ovarian GCs, with the process being essential for improving follicular growth and reproductive performance in ewes [19]. Dietary supplementation of 0.07% NAC has further been found to regulate endometrial immune and inflammatory responses through the PI3K/Akt and PPAR pathways, hence improving the survival rate of goat embryos in early pregnancy [20]. In fact, previous studies have specifically emphasized the pivotal role of the PI3K/Akt pathway in governing the growth and apoptosis of GCs throughout follicular maturation [21,22,23,24]. However, despite the above insights, the exact mechanism of NAC in female rabbits remains relatively less explored.

To address this gap, an in vitro culture model of GCs from New Zealand female rabbits was used to assess the impact of NAC on GC proliferation and apoptosis, especially after D-gal-induced damage to these cells. In addition, RNA sequencing technology was applied to identify genes with altered expression following exposure to D-gal and NAC. Finally, bioinformatics methods were used to further analyze the genetic pathways affected in GCs following the addition of these substances. This research aims to provide a comprehensive theoretical understanding of the molecular-level regulatory actions of NAC which are relevant to the reproductive functions of female rabbits.

## 2. Materials and Methods

### 2.1. Animal and Sample Collection

In this study, healthy and sexually mature 6-month-old New Zealand female rabbits of comparable body weights were selected. Fresh ovarian tissues were harvested from the animals and after sterilization with 75% ethanol, the tissues were meticulously washed three times using a bacteriostatic phosphate-buffered saline (PBS) solution (enriched with an antibiotic cocktail containing 100 U/mL penicillin and 0.1 mg/mL streptomycin and purchased from Beyotime, Shanghai, China). Then the ovaries were immersed in DMEM/F12 medium supplemented with penicillin and streptomycin, and follicles were punctured with syringe needles to collect cell fluid, which was subsequently filtered and centrifuged before discarding the supernatant. Similarly, follicles 2.5–5 mm in diameter were punctured to collect the follicular fluid. The liquid medium mixture was then centrifuged at 1000 revolutions per min for 5 min, after which the supernatant was discarded. The resulting cells were resuspended in DMEM/F12 medium, enriched with 10% fetal bovine serum (Gibco, Carlsbad, CA, USA), and they were subsequently seeded into culture dishes prior to incubation for 24 h in a thermostat-regulated incubator maintained at 37 °C and under a controlled 5% CO_2_ atmosphere. Animal management and all experimental methods were stringently implemented in accordance with the ethical standards and protocols approved by the Animal Care and Use Committee of Yangzhou University.

### 2.2. Culture of Granulosa Cells

To assess NAC’s impact on D-gal-induced GC injury, different concentrations of D-gal (0.5, 1 and 1.5 mg/mL) were first used to establish a GC injury model. Varying doses of NAC (0.5, 1, and 1.5 mg/mL) were then added to the cells for 12 h to observe the effects of NAC.

### 2.3. Quantitative Real-Time PCR

Total RNA, extracted from the cells using the SteadyPure RNA Extraction Kit (AG21024) (ACCURATE BIOTECHNOLOGY (HUNAN) Co., Ltd., Changsha, China), was reverse transcribed into cDNA with the HiScript II Q Select RT SuperMix (Vazyme Biotech Co., Ltd., Nanjing, China). Gene expression levels were subsequently quantified by real-time PCR, performed on a QuantStudio^®^ 5 system (Applied Biosystems, Thermo Fisher Scientific, Waltham, MA, USA) using the AceQ qPCR SYBR^®^ Green Master Mix (Vazyme Biotech Co., Ltd., Nanjing, China). The relative gene expression was eventually determined based on the 2^−∆∆Ct^ method [25], with glyceraldehyde 3-phosphate dehydrogenase (GAPDH) acting as an internal reference control. The specific primer sequences designed for each gene are provided in Appendix A.

### 2.4. Cell Apoptosis and Proliferation Assays

Rabbit ovarian GCs were seeded into 24-well culture plates at a density of 2.0 × 10^5^ cells per well, after which cell apoptosis was determined using the Annexin V-FITC Apoptosis Detection Kit (Vazyme, Nanjing, China) according to the manufacturer’s instructions. This was followed by fluorescence-activated cell sorting analysis of the apoptotic cells, performed by flow cytometry on a FACSAria SORP system (Becton Dickinson, San Diego, CA, USA). To assess cell proliferation, the CCK8 assay kit (Vazyme, Nanjing, China) was used in accordance with the provided instructions. Briefly, cells were seeded into 96-well plates at a concentration of 1.0 × 10^4^ cells per well and treated with D-gal and NAC as required for each experimental group. Absorbance values were subsequently measured at a wavelength of 450 nm using a high-performance multimode microplate reader (Infinite™ M200 PRO, Tecan, Grödig, Austria).

### 2.5. Protein Extraction and Western Blotting

Cell samples were lysed in RIPA buffer (PPLYGEN, Beijing, China) to extract proteins, which were subsequently quantified using an Enhanced BCA Protein Assay Kit (Beyotime, Shanghai, China). All protein samples were then standardized to a concentration of 0.5 µg/µL, with each well receiving 3 µL of this adjusted solution. This was followed by automated protein separation achieved with the use of a Western blotting system (ProteinSimple, San Jose, CA, USA), as described by Harris [26]. The antibodies used in this set of experiments are listed in Appendix A.

### 2.6. Immunofluorescence (IF) Staining

Cells from the control group, the 1.5 mg/mL D-gal-treated group, and the 0.5 mg/mL NAC-treated group were seeded into 24-well plates and allowed to grow until they reached a 50–60% confluent state. The culture medium was then discarded, and after washing the cells three times with PBS, they were fixed in 4% paraformaldehyde solution at room temperature for 30 min. This was followed by cell permeabilization using 0.3% TritonX-100 for 1 h and subsequent blocking with 1% bovine serum albumin (BSA) at room temperature for another hour.

The cells were then incubated overnight at 4 °C with the primary antibodies Bax (diluted at 1:500 ratio) and BCL-2 (diluted at 1:500) for a total duration of 12 to 16 h. This was followed by a 1 h incubation with secondary antibodies (goat anti-mouse IgG and goat anti-rabbit IgG) at a temperature of 37 °C, after which the nuclei were stained with 4′,6-diamidino-2-phenylindole (DAPI). The state of the GCs was eventually assessed under a fluorescence microscope (OLYMPUS).

### 2.7. Intracellular Reactive Oxygen Species (ROS)

After treating GCs with 1.5 mg/mL of D-gal and 0.5 mg/mL of NAC, the cells were incubated for 20 min at 37 °C in a solution containing 10 mM of DCFH-DA. They were subsequently rinsed three times using serum-free cell culture medium to ensure complete removal of extracellular DCFH-DA. The intracellular ROS levels were finally detected by flow cytometry analysis.

### 2.8. Detection of Antioxidative Enzymes

The enzymatic activities of catalase (CAT), superoxide dismutase (SOD), and glutathione (GSH) were quantified using commercially available assay kits (Jiancheng, Nanjing, China) according to the manufacturer’s instructions.

### 2.9. RNA-Seq and Bioinformatics Analysis

Total RNA, extracted from the three groups of cells (control, 1.5 mg/mL D-gal-treated, and 0.5 mg/mL NAC-treated groups), were sent to Biomarker Technologies Co., Ltd. (Beijing, China) for constructing cDNA libraries, sequencing as well as processing the resulting transcriptome data. For each experimental condition, there were three independent biological replicates.

Functional enrichment and pathway analyses were then performed on differentially expressed genes (DEGs) with the R Bioconductor’s clusterProfiler package to explore Gene Ontology (GO) terms and Kyoto Encyclopedia of Genes and Genomes (KEGG) pathways. In this case, the DESeq2 version 1.30.1 was used to identify significant variations in gene expression between the three experimental groups, with the criteria to establish a gene as a DEG being a log2(FoldChange) of ≥1 and a *p*-value of <0.05.

### 2.10. Statistical Analysis

Each experimental procedure was replicated a minimum of three times to ensure consistency, with all collected data subsequently expressed as mean ± SEM (standard error of the mean). The different results from RT-PCR, cell apoptosis assays, and proliferation experiments were then statistically analyzed using either a two-sided Student’s *t*-test or one-way analysis of variance (ANOVA) in SPSS 25.0 software (SPSS Inc., Chicago, IL, USA). In addition, graphical illustrations were generated using GraphPad Prism 8 software (GraphPad Software, Inc., San Diego, CA, USA). Results were considered to be statistically significant at *p* < 0.05 (*) and of higher significance at *p* < 0.01 (**).

## 3. Results

### 3.1. Construction of D-Gal-Induced Ovarian GC Injury Model

Primary GCs from rabbits were cultured, and once the cell confluence reached 80–90%, they were exposed to increasing doses of D-gal for 12 h. In the control group, the cell fusion degree was high, the cell spread was very open, the size was uniform, the shape was complete, the cells were closely connected, and the boundary was clear. However, in the D-gal group, the number of surviving cells decreased, the number of atypical cells increased, and the cell debris in the culture medium increased, which was more obvious with the increase in D-gal concentration (Figure 1B). Results of CCK-8 assays, performed to evaluate the cytotoxic effects of D-gal, subsequently showed that higher concentrations induced a significant and proportional decline in cell proliferation (*p* < 0.01). Specifically, at 1.5 mg/mL of D-gal, cell viability dropped by approximately 50% (IC50: 1.606 mg/mL) (Figure 1C,D), with a decrease in the expression of the PCNA (proliferating cell nuclear antigen) protein also noted at this concentration (*p* < 0.01) (Figure 1E). In addition, flow cytometry analysis showed that the apoptosis rate of GCs treated with 1.5 mg/mL D-gal was the highest, reaching 19.62% (Figure 1F).

Additional analysis revealed that the mRNA and protein expression levels of Bax/BCL-2, Caspase-9, as well as p53 genes were significantly higher across all tested concentrations of D-gal (*p* < 0.05 or *p* < 0.01), as depicted in Figure 1G,H. These findings collectively demonstrated that D-gal could induce apoptosis in rabbit ovarian GCs in a dose-dependent fashion, with optimal effects observed at a concentration of 1.5 mg/mL.

### 3.2. NAC Decreased D-Gal-Induced Apoptosis and Increased Cell Viability in Ovarian GCs

After establishing the D-gal-induced ovarian GC injury model, the cells were exposed to different concentrations of NAC for a 12 h treatment. In this case, CCK-8 and PCNA assays revealed that a concentration of 0.5 mg/mL could significantly enhance GC proliferation. However, this effect was reduced as the concentration of NAC increased, with cell proliferation being eventually suppressed at 1.5 mg/mL (*p* < 0.05, *p* < 0.01) (Figure 2A–C).

Subsequent assays further revealed that 0.5 mg/mL of NAC could significantly reduce cell apoptosis (*p* < 0.01) (Figure 2D). Additionally, qPCR and WB analyses demonstrated that, at this concentration, NAC significantly decreased the mRNA and protein expression levels of Bax/BCL-2, Caspase-9, and p53 compared with the control group, although these levels still increased at higher NAC concentrations (*p* < 0.05 or *p* < 0.01) (Figure 2E,F). Hence, it was speculated that 0.5 mg/mL of NAC was suitable to alleviate D-gal-induced apoptosis in GCs while stimulating proliferation.

### 3.3. NAC Could Alleviate D-Gal-Induced Oxidative Stress in GCs

To further probe the potential regulatory functions of NAC, its capacity to alleviate D-gal-induced oxidative stress in GCs was examined.

As shown in Figure 3A, there was a significant decrease in the cellular levels of CAT, GSH, and SOD enzymes after exposure to 1.5 mg/mL of D-gal compared with the untreated control group (*p* < 0.05). In contrast, treatment with 0.5 mg/mL NAC led to a significant increase in the levels of these antioxidant enzymes compared with the 1.5 mg/mL D-gal-treated group (*p* < 0.05). In addition, the findings of this study revealed increased ROS levels, along with a significant increase in the release of Cyt c within the 1.5 mg/mL D-gal-treated group (*p* < 0.05). However, on adding 0.5 mg/mL of NAC, the ROS levels decreased significantly, and this was reflected in a corresponding decrease in the amount of released mitochondrial Cyt c (*p* < 0.05) (Figure 3B–D).

Overall, 1.5 mg/mL of D-gal could increase the ROS level in GCs, with mitochondria being the primary target for the pro-apoptotic effects of excessive ROS. The latter subsequently induced oxidative stress within cells, causing a rupture in the mitochondrial outer membrane and subsequent apoptosis. However, following supplementation with 0.5 mg/mL of NAC, ROS levels decreased, along with a lower release of mitochondrial Cyt c as well as less apoptotic events. Thus, an appropriate dose of NAC could potentially alleviate D-gal-induced oxidative stress in GCs and mitigate apoptosis through the mitochondrial pathway.

### 3.4. RNA-Seq Analysis of NAC’s Effect on Rabbit Ovarian GCs

To validate the study’s hypothesis and further explore the mechanism of NAC’s action on D-gal-induced apoptosis in GCs, RNA sequencing analysis was performed on GCs from the control, 1.5 mg/mL D-gal-treated, and 0.5 mg/mL NAC-treated groups. The results of the volcano plot highlighted the upregulation and downregulation of DEGs after D-gal and NAC treatments, compared with the control group (Figure 4A). Moreover, a Venn diagram illustrated significant differences between the expression levels of 532 genes across the three groups (Figure 4B). GO enrichment analyses suggested that both D-gal and NAC treatments induced stress responses in cells, thereby impacting cell growth cycle processes (Figure 4C). Furthermore, KEGG pathway enrichment analysis indicated that the PI3K/Akt/mTOR signaling pathway was significantly upregulated after administering D-gal. In addition, it also included TNF, Focal adhesion, apoptosis, and other signaling pathways (Figure 4D). Based on the results, six differentially expressed genes were randomly identified. The findings from the qPCR analysis corroborated those of the RNA sequencing, thereby validating the reliability of the sequencing data (Figure 4E,F).

### 3.5. NAC Inhibited D-Gal-Induced GC Apoptosis through the PI3K/Akt/mTOR Pathway

The PI3K/Akt/mTOR signaling cascade plays a crucial role in regulating apoptosis. To explore whether NAC could modulate D-gal-induced apoptosis in GCs through this pathway, the protein expression was scrutinized. WB analysis revealed that, compared with the controls, the ratios of p-PI3K/PI3K, p-AKT/AKT, and p-mTOR/mTOR were significantly reduced in GCs treated with 1.5 mg/mL of D-gal (*p* < 0.01). Conversely, treating the cells with 0.5 mg/mL of NAC led to a significant increase in these phosphorylation ratios (*p* < 0.05) (Figure 5A). Additionally, immunofluorescence (IF) staining showed that the fluorescence intensity of Bax was significantly heightened, while that of BCL-2 decreased in the 1.5 mg/mL D-gal-treated group compared with the control. When 0.5 mg/mL of NAC was added, Bax’s signal intensity lessened, while that of BCL-2 increased (Figure 5B).

Therefore, exposure to 1.5 mg/mL of D-gal resulted in a significant increase in the phosphorylation status of PI3K, Akt, and mTOR, all of which occurred simultaneously along with increased expression of Bax and decreased expression of BCL-2. However, supplementation with 0.5 mg/mL of NAC significantly attenuated the D-gal-induced phosphorylation. These results indicated that NAC potentially mitigates D-gal-induced harm to ovarian GCs in New Zealand female rabbits through the modulation of the PI3K/Akt/mTOR signaling pathway.

## 4. Discussion

D-gal is widely recognized as an aging agent that can stably replicate the systemic aging process in animal models. At the same time, it possesses a number of advantages, including simple operation, obvious modeling, moderate cycle, and low cost [27]. D-gal has largely been used to simulate normal organ aging or oxidation processes in different tissues, with the latter mainly manifested as oxidative damage, decreased cell proliferation, as well as abnormal gene expression and regulation, just to name a few. For example, in order to verify the antioxidant effects of Ginsenoside Rg1, D-gal was injected into mice to induce oxidative stress in vitro [28]. Similarly, a D-gal-induced ovarian aging model was established to evaluate the protective effects of lycopene on ovarian oxidative stress during ovarian aging [29], while in a different study, an ethanolic extract of propolis was used to determine the effect of D-gal on the injury of mouse C2C12 cells [30]. NAC is an effective antioxidant that can directly react with oxidative metabolites to improve cell viability and regulate the cell cycle and apoptosis [31,32,33]. As such, it is widely used in medicine to protect cells from oxidative damage. In this context, studies have shown that long-term use of NAC could actually increase the number of mouse oocytes and improve their quality [14]; specifically, low NAC concentrations (0.1 μM and 1 μM) were shown to promote the proliferation of porcine placental trophoblast cells [12], while the addition of 1.5 mM of NAC for 24 h could significantly promote the development of porcine oocytes and blastocysts [34]. Thus, by reducing the level of oxidative stress and apoptosis of GCs, NAC can promote the proliferation of GCs in mice, restore hormone levels, and improve follicular development [13]. However, the precise molecular underpinnings of NAC’s impact on rabbit GCs remain poorly understood. Consequently, this study aimed to elucidate the detailed molecular mechanism through which NAC influences D-gal-induced damage to ovarian GCs in rabbits.

Apoptosis represents a basic physiological mechanism for maintaining the stability of organisms. GCs are the largest cell group of follicles and participate in many physiological processes of the ovary, including follicular growth, luteal lysis, premature ovarian dysfunction, and multiple ovarian syndromes [35,36]. In this context, follicular atresia represents one of the main factors causing a decline in the reproductive capacity of livestock, with GC apoptosis regulating the growth and atresia of follicles [5,37]. The PI3K/Akt/mTOR signaling cascade has a critical function in the regulation of cellular growth, proliferation, and the production of proteins [38]. It also facilitates the transcription and translation of anti-apoptotic gene proteins, such as those belonging to the BCL-2 gene family, thereby playing a pivotal role in enhancing GC proliferation and suppressing GC apoptosis [39,40,41,42]. However, it remains unclear whether NAC affects the PI3K/Akt/mTOR pathway in rabbit GCs.

In this work, the impact of D-gal on the proliferation and apoptosis of GCs was assessed, with the results showing that the effects were dose-dependent. Specifically, at a concentration of 1.5 mg/mL, D-gal could effectively suppress the growth of GCs while concurrently accelerating their programmed cell death. On this basis, different concentrations of NAC were added, and it was found that low concentrations (0.5 mg/mL) could significantly inhibit the apoptosis of GCs and promote their proliferation. However, it was also observed that at a high enough concentration of NAC (1.5 mg/mL), increased apoptosis and inhibited proliferation could be observed in rabbit GCs. This outcome might be attributed to the fact that NAC is a mild organic acid; when its concentration within the medium becomes excessively elevated, it can cause a drop in pH levels, thereby damaging the cell membrane, a phenomenon that has been documented in several previous cellular research studies [12,43].

In this study, the different proteins participating in relevant signaling pathways were identified before further exploring the underlying mechanism through which NAC exerts its protective effects on GCs. The findings revealed that, in comparison with the control group, the ratios of p-PI3K to total PI3K, p-Akt to total Akt, and p-mTOR to total mTOR were significantly reduced in the 1.5 mg/mL D-gala-treated group. This could have been due to the phosphorylation of PI3K after adding D-gal, thus establishing PI3K as being biologically active. Upon activation, PI3K stimulates the phosphorylation of Insulin Receptor Substrate-1 (IRS-1) which, in turn, generates PIP3. The latter then selectively interacts with and binds Akt along with 3-Phosphoinositide-Dependent Protein Kinase 1 (PDK1), thereby leading to Akt activation. The activated Akt can then catalyze the phosphorylation of a variety of downstream effector proteins. In the aging-related process, the change in Akt activity and its phosphorylation of downstream anti-apoptotic protein BCL-2 family members are particularly significant. Akt can inhibit some pro-apoptotic members of the BCL-2 family, such as Bax, by phosphorylation, thereby reducing its ability to promote cell death, maintaining cell survival and participating in cell cycle regulation. At the same time, Akt also affects mTOR, the core controller of cell growth and metabolism. Akt can promote the activation of mTOR complex 1 (mTORC1) through direct or indirect pathways, thereby regulating key cellular processes such as protein synthesis, autophagy, and cell size. The highly active mTOR pathway is often associated with cell proliferation, metabolism, and apoptosis, but long-term overactivation may also lead to excessive resource consumption and accelerate aging and disease. This cascade of events ultimately helps to regulate cell proliferation and apoptosis [44,45,46,47,48]. However, 0.5 mg/mL NAC can significantly increase the ratio of p-PI3K/PI3K, p-Akt/Akt, and p-mTOR/mTOR, indicating that NAC can reduce the phosphorylation of PI3K, Akt, and mTOR to reduce D-gal-induced damage.

Mitochondria are highly abundant in GCs, providing sufficient energy for cell proliferation and differentiation [49]. ROS, which primarily originate from the byproducts of mitochondrial metabolism, have a dual impact on cells, being essential for normal cell functioning while also forming the basis for oxidative damage. In the latter case, excessive ROS can induce permeabilization of the mitochondrial outer membrane, thereby releasing calcium ions and Cytochrome c (Cyt c), which eventually lead to apoptosis. CAT enzymes are ubiquitously present within cells and help to shield them from the detrimental effects of ROS, thereby preserving normal cellular functions. GSH, on the other hand, represents a primary reservoir of sulfhydryl groups in most biological cells. As an important reducing agent, it participates in various redox reactions in vivo. Finally, SOD plays a critical part in maintaining the equilibrium between oxidative and antioxidative processes. In this study, it was found that the ROS levels in GCs increased significantly after adding 1.5 mg/mL of D-gal. At the same time, this resulted in significantly reduced SOD, CAT, and GSH activities, along with an increasing release of Cyt c from mitochondria. It has also been reported that ROS can directly activate mitochondrial permeability transition, leading to the release of mitochondrial Cyt c [50]. However, when 0.5 mg/mL of NAC was added, the ROS level of GCs decreased significantly, while the activities of the SOD, CAT, and GSH enzymes also increased significantly. Mitochondria reduced the release of Cyt c, which proved that NAC could help the body remove excessive ROS and protect cells from oxidative stress.

## 5. Conclusions

In this study, an experimental model of D-gal-induced rabbit ovarian GCs was established to explore the protective effects of NAC on GC injury as well as to elucidate its underlying mechanism. NAC alleviated D-gal-induced GC damage through the PI3K/Akt/mTOR signaling pathway. In addition, NAC enhanced the activity of various antioxidant enzymes (SOD, CAT, and GSH) before repairing the damaged GCs to restore secretory functions. In conclusion, NAC mitigates D-gal-induced harm to ovarian GCs in New Zealand female rabbits through modulation of the PI3K/Akt/mTOR pathway. This research provides the groundwork for future studies exploring the impact of NAC on animal reproductive function at the cellular level.

## Figures and Tables

**Figure 1 antioxidants-13-00384-f001:**
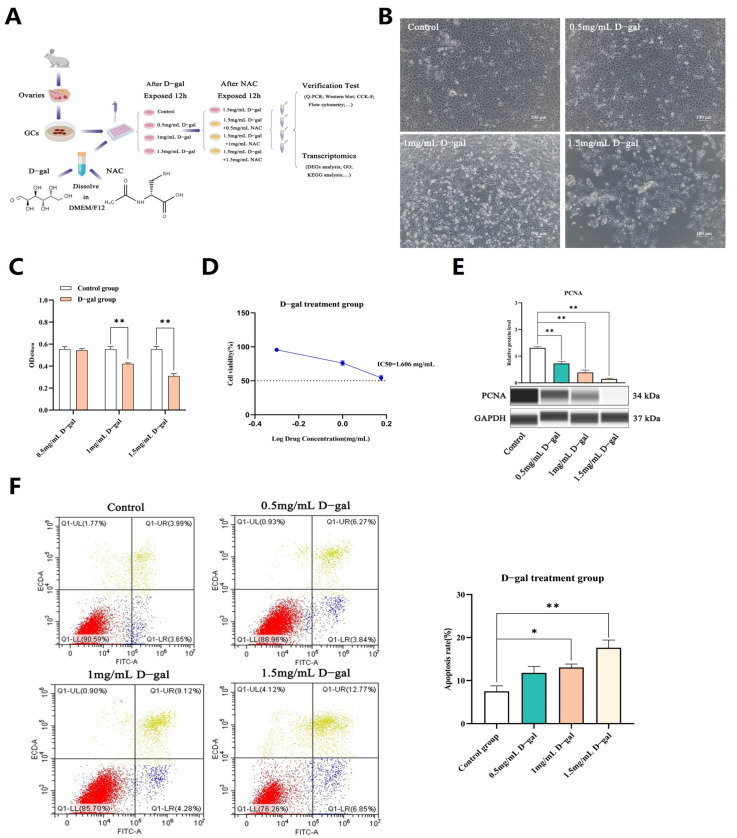
Construction of a D-gal-induced ovarian GC injury model. (**A**) The experimental process to study the model of ovarian GCs in vitro. (**B**) Morphological changes in GCs following a 12 h culture with varying doses of D-gal in vitro. (**C**) The CCK-8 assay was used to assess the effects of varying D-gal concentrations on the proliferation of GCs. (**D**) Calculation of IC50 values representing the drug concentration which causes a 50% reduction in cell proliferation. (**E**) WB analysis was performed to evaluate the expression levels of the PCNA protein in GCs following a 12 h exposure to varying concentrations of D-gal. (**F**) The rate of apoptosis in GCs was quantified by flow cytometry following a 12 h exposure to different concentrations of D-gal. (**G**) qRT-PCR was used to assess the expression levels of apoptosis-related genes in GCs after treatment with 0.5, 1, and 1.5 mg/mL of D-gal. (**H**) WB analysis was performed to examine the expression levels of apoptosis-related proteins in GCs after treatment with 0.5, 1, and 1.5 mg/mL of D-gal. All data are presented as mean ± SEM and were statistically analyzed with paired two-tailed *t*-tests at * *p* < 0.05 and ** *p* < 0.01.

**Figure 2 antioxidants-13-00384-f002:**
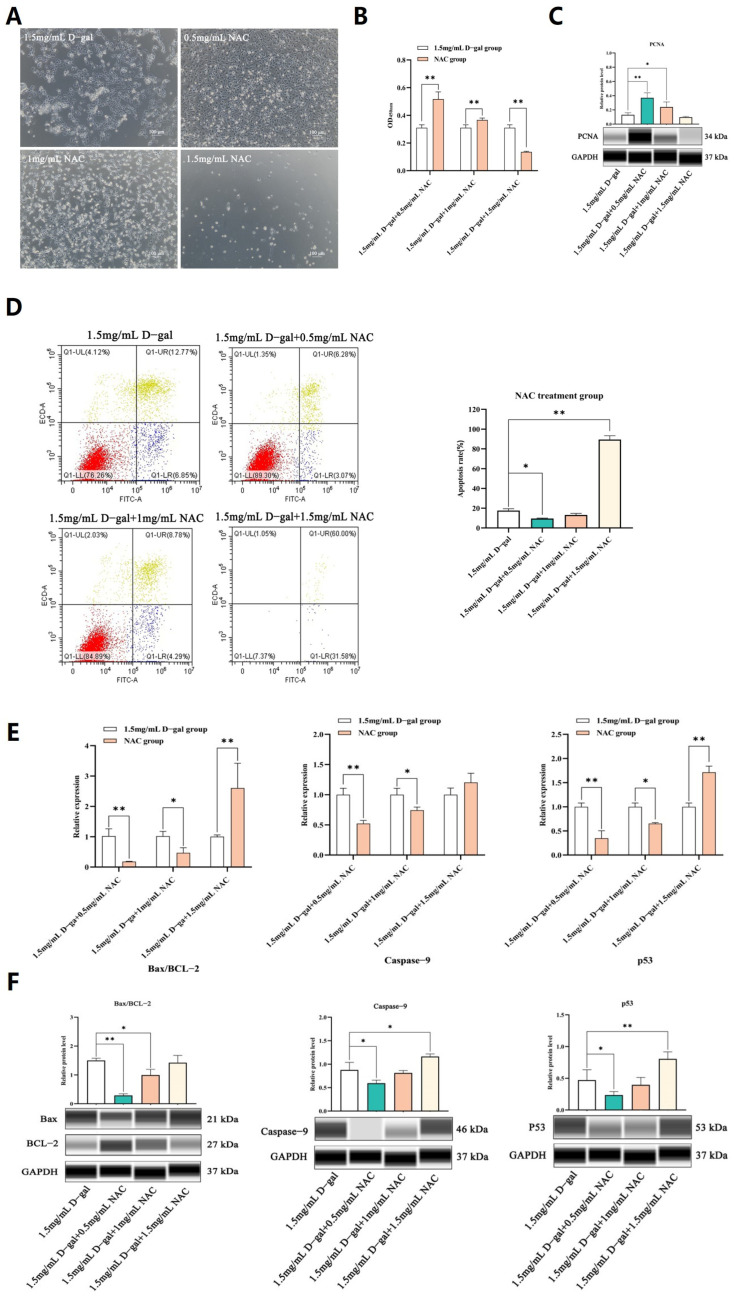
NAC decreased D-gal-induced apoptosis and increased cell viability in ovarian GCs. (**A**) Morphological changes in GCs after exposure to different concentrations of NAC for 12 h in vitro. (**B**) The CCK-8 assay was used to assess the proliferation rate of GCs when exposed to different doses of the NAC treatment. (**C**) The expression levels of the PCNA protein in GCs, treated with different concentrations of NAC for 12 h, as determined by WB. (**D**) The apoptosis rate of GCs was measured after a 12 h incubation with different levels of NAC using Annexin V-FITC/PI staining and subsequent flow cytometry analysis. (**E**) The expression levels of apoptosis-related genes in GCs treated with 0.5, 1, and 1.5 mg/mL of NAC were assessed through qRT-PCR. (**F**) The expression of apoptosis-related genes in GCs treated with 0.5, 1, and 1.5 mg/mL NAC was then analyzed by WB. All data are presented as mean ± SEM and the results were statistically analyzed with paired two-tailed *t*-tests at * *p* < 0.05 and ** *p* < 0.01.

**Figure 3 antioxidants-13-00384-f003:**
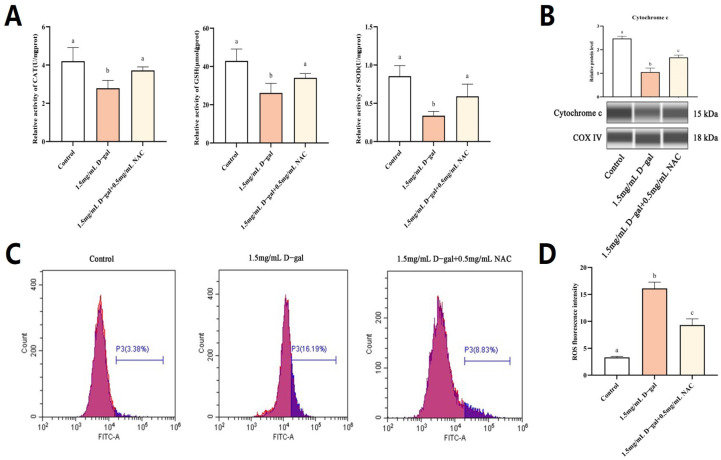
NAC can alleviate D-gal-induced oxidative stress in GCs. (**A**) Enzymatic assays revealed the activities of CAT, GSH, and SOD within GCs. (**B**) The protein expression level of Cyt c in mitochondria was assessed, with Cytochrome c oxidase IV (COX IV) serving as an internal control for the mitochondrial content. (**C**,**D**) Flow cytometry was then applied to detect changes in intracellular levels of ROS after treatment with 1.5 mg/mL of D-gal and 0.5 mg/mL of NAC. Data are presented as mean ± SEM; distinct lowercase letters (a, b, c) denote significant differences among groups, while identical letters imply the absence of a statistically significant difference. Statistical analysis was conducted using a paired two-tailed *t*-test.

**Figure 4 antioxidants-13-00384-f004:**
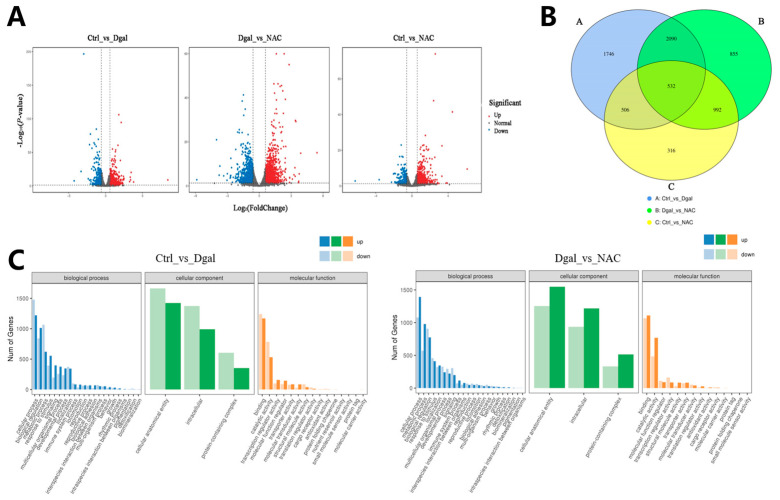
RNA-Seq analysis of NAC’s effect on rabbit ovarian GCs. (**A**) Volcano plot of DEGs. (**B**) Venn diagram illustrating DEG overlap between the control, 1.5 mg/mL D-gal-treated, and 0.5 mg/mL NAC-treated groups. (**C**) GO enrichment statistics were visualized as histogram plots where the horizontal axis represented GO classifications, the vertical axis showed gene counts, and the various colors represented different primary categories. (**D**) Dot plot presenting the top 20 enriched KEGG pathways; each dot symbolized a pathway with its name on the *y*-axis and enrichment factor on the *x*-axis. (**E**) The expression levels of *CDCA7*, *MSRB1*, *PAICS*, *THRB*, *ATF3,* and *KLF12* genes at the time of sequencing. (**F**) The expression levels of *CDCA7*, *MSRB1*, *PAICS*, *THRB*, *ATF3,* and *KLF12* genes were detected by qPCR. Data are presented as mean ± SEM; distinct lowercase letters (a, b, c) denote significant differences between groups, while identical letters imply the absence of a statistically significant difference. Statistical analysis was conducted using a paired two-tailed *t*-test.

**Figure 5 antioxidants-13-00384-f005:**
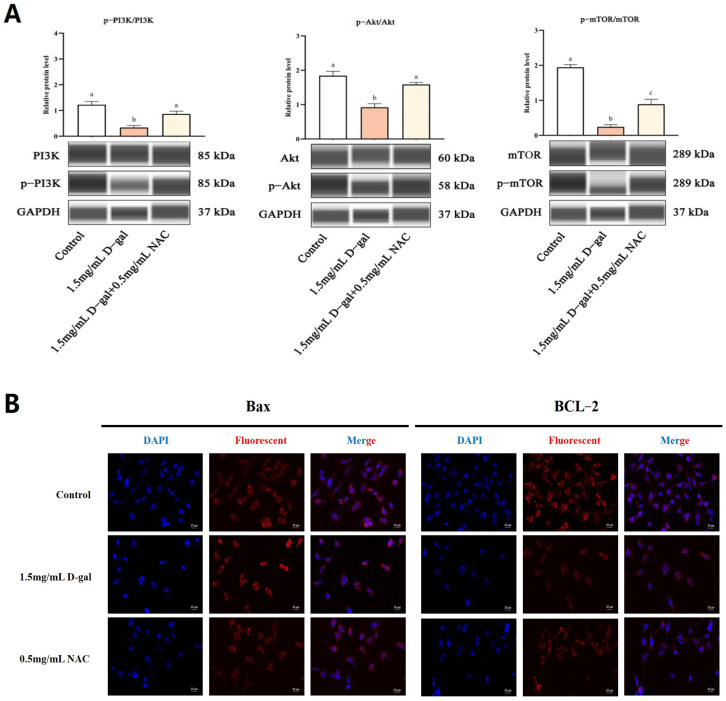
NAC inhibited D-gal-induced GC apoptosis through the PI3K/Akt/mTOR pathway. (**A**) WB determined the expression levels of p-PI3K, PI3K, p-AKT, AKT, p-mTOR, and mTOR. (**B**) IF was used to detect the expression of *Bax* and *BCL-2* genes in the control, 1.5 mg/mL D-gal-treated, and 0.5 mg/mL NAC-treated groups. Scale bar: 20 μm. Data are presented as mean ± SEM; distinct lowercase letters (a, b, c) denote significant differences between groups, while identical letters imply the absence of statistically significant differences. Statistical analysis was performed using paired two-tailed *t*-tests.

## Data Availability

All figures and tables used to support the results of this study are included.

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
