# Peer review of "N-Acetylcysteine Alleviates D-Galactose-Induced Injury of Ovarian Granulosa Cells in Female Rabbits by Regulating the PI3K/Akt/mTOR Signaling Pathway"

_antioxidants, 2024, doi:10.3390/antiox13040384_

Round 1

Reviewer 1 Report

Dear authors,

The topic of the paper is really interesting about the role of NAC in the regulation of PI3K/Akt/mTOR in the damage in ovarian granulosa cells. However, there are some comments, the authors should address:

Comment 1. In the list of antibodies, the authors should add the reference and commercial company of the antibodies.

Comment 2. In the section of results 3.1, the authors should add one description of the morphological changes in the cells after the treatment with D-Gal.

Comment 3. In the Figure F, the authors should indicate which parameter of FACs indicate Annexin V levels

Comment 4. The quality of graphs is poor. They should improve it.

I think the details in tables, figures is clear.

Reviewer 2 Report

The objective of this study was to examine the effects of NAC on ovarian damage in female rabbits. For this purpose, D-galactose (D-gal) was first used to establish a model of damaged granulosa cells, with exposure to 1.5 mg/mL of D-gal leading to substantial damage. Subsequently, varying concentrations of NAC were introduced to determine the precise mechanism through which it influences cell damage. The scientific investigation is very interesting, however, some minor problems, as indicated below, should be addressed before the document can be considered for publication in this journal. This version of the manuscript is not enough complete.

Here, I present all my objections in details.

Minor revision:

 English language and style are fine, minor spell check is required to ensure that an international audience can clearly understand your text.

Line 12. The authors should explain better the following terminology: “economic potential”

In the abstract, the authors should summarizes the results obtained.

Line 51. The authors should indicate used concentration.

Line 53.  The authors should explain the underlying aging mechanisms. 

Line 94. What are used antibiotics?

Line 10. What is the incubation time?

Line 147. The authors should modify ambient with room temperature.

The authors should improve the resolution of Figure 2 A.

ROS, which primarily originate from the byproducts of mitochondrial metabolism, have a dual impact on cells, being essential for normal cell functioning while also forming the basis for oxidative damage. In the latter case, excessive ROS can induce permeabilization of the mitochondrial outer membrane, thereby releasing calcium ions and Cytochrome c (Cyt c) which eventually lead to apoptosis.

Overall, 1.5 mg/mL of D-gal could increase the ROS level in GCs, with mitochondria being the primary target for the pro-apoptotic effects of excessive ROS. The latter subsequently induce oxidative stress within cells, causing a rupture in the mitochondrial outer membrane and subsequent apoptosis. However, following supplementation with 0.5 mg/mL of NAC, ROS levels decreased, along with a lower release of mitochondrial Cyt c as well as less apoptotic events. Thus, an appropriate dose of NAC could potentially alleviate D-gal-induced oxidative stress in GCs and mitigate apoptosis through the mitochondrial pathway.

These sentences should be included in the discussion section.

Line 394. The authors should explain better the phosphorylation mechanisms related to aging processes.

No detail comments are necessary 
